# The Visual Patient Avatar ICU Facilitates Information Transfer of Written Information by Visualization: A Multicenter Comparative Eye-Tracking Study

**DOI:** 10.3390/diagnostics13223432

**Published:** 2023-11-12

**Authors:** Julie Viautour, Lukas Naegeli, Julia Braun, Lisa Bergauer, Tadzio R. Roche, David W. Tscholl, Samira Akbas

**Affiliations:** 1Institute of Anesthesiology, University Hospital Zurich, Raemistrasse 100, 8091 Zurich, Switzerland; julie.viautour@usz.ch (J.V.); lisa.bergauer@gmx.de (L.B.); tadzioraoul.roche@usz.ch (T.R.R.); david.tscholl@usz.ch (D.W.T.); 2Master Program in Biostatistics, Epidemiology, Biostatistics and Prevention Institute, University of Zurich, Hirschengraben 84, 8001 Zurich, Switzerland; lukas.naegeli2@uzh.ch; 3Departments of Epidemiology and Biostatistics, Epidemiology, Biostatistics and Prevention Institute, University of Zurich, Hirschengraben 84, 8001 Zurich, Switzerland; julia.braun@uzh.ch

**Keywords:** Visual-Patient-avatar ICU, eye tracking, visual perception, situation awareness, avatar-based monitoring

## Abstract

Patient monitoring is crucial in critical care medicine. Perceiving and interpreting multiple vital signs requires a high workload that can lead to decreased situation awareness and consequently inattentional blindness, defined as impaired perception of unexpectedly changing data. To facilitate information transfer, we developed and validated the Visual-Patient avatar. Generated by numerical data, the animation displays the status of vital signs and patient installations according to a user-centered design to improve situation awareness. As a surrogate parameter for information transfer in patient monitoring, we recorded visual attention using eye-tracking data. In this computer-based study, we compared the correlation of visually perceived and correctly interpreted vital signs between a Visual-Patient-avatar ICU and conventional patient monitoring. A total of 50 recruited study participants (25 nurses, 25 physicians) from five European study centers completed five randomized scenarios in both modalities. Using a stationary eye tracker as the primary endpoint, we recorded how long different areas of interest of the two monitoring modalities were viewed. In addition, we tested for a possible association between the length of time an area of interest was viewed and the correctness of the corresponding question. With the conventional monitor, participants looked at the installation site the longest (median 2.13–2.51 s). With the Visual-Patient-avatar ICU, gaze distribution was balanced; no area of interest was viewed for particularly long. For both modalities, the longer an area was viewed, the more likely the associated question was answered incorrectly (OR 0.97, 95% CI 0.95–0.99, *p* = 0.008). The Visual-Patient-avatar ICU facilitates and improves information transfer through its visualizations, especially with written information. The longer an area of interest was viewed, the more likely the associated question was answered incorrectly.

## 1. Introduction

Continuous patient monitoring is highly accepted by intensive care personnel, supporting them in daily patient surveillance and making critical treatment decisions [1]. Although patient monitoring has become an integral part of the daily routine of health care providers and increases patient safety, the increasing volume of data presented leads to a high perceptual load, which impairs the ability to detect an unexpected change in the data [2,3,4,5]. This effect, known as inattentional blindness, is particularly aggravated when many different parameters with similar values are displayed, e.g., heart rate and saturation both being able to have a number between 60 and 100 [6,7]. Together with the increasing workload due to the care of several critically ill patients under time pressure, this may lead to medical errors with a possible negative outcome for patients [8,9,10].

Previous studies showed that conventional patient monitoring, based on numbers and curves, is not ideal for conveying patient information [11,12]. In addition, intensive care personnel spend only 3% of their working time looking at the patient monitor, which adds up to just under 15 min in an eight-hour workday [13]. Therefore, it has been recommended in the past to develop new technologies that improve the transfer of information [11].

The principle of situation awareness, according to Endsley et al., consists of three parts: “the perception of elements of the environment within a volume of time and space, the comprehension of their meaning, and the projection of their status into the near future” with errors occurring most frequently in the area of perception [3,4,14,15]. To facilitate information transfer and thus improve situation awareness, we developed and validated the Visual-Patient avatar in several computer-based studies, a high-fidelity simulation, and eye-tracking studies [16,17,18,19,20,21]. The Visual-Patient avatar preprocesses and displays vital signs according to a user-centered design and principles of logic, so that, for example, the heart rate is represented as a pulsation of the avatar. In the recently developed ICU version, in addition to the vital parameters, the present installations are also displayed [4,22,23]. The installations, including central venous or arterial catheters, are presented at their corresponding location on the avatar. To the best of our knowledge, the display of patient installations on the conventional monitor has not been common practice to date.

The visual interaction of the health care providers with the patient monitors can be objectified by eye tracking. The resulting data allow a deeper understanding of underlying analytical pathways by providing information about spatial and temporal measurements, gaze coordinates, dwell times on areas of interest (AOI), as well as saccades and fixations [24,25]. Since information transfer in the setting of patient monitoring is defined by visual attention, the information obtained could be used as a surrogate parameter for situation awareness, allowing the workload and important insights to be derived from it [26,27,28,29].

In this study, we compared the Visual-Patient-avatar ICU with the conventional monitor using different scenarios with regard to the detection of the status of individual vital signs and the present installations. We hypothesize that Visual-Patient-avatar ICU facilitates information transfer through animated visualizations and that the correlation of the observed parameters with the correctly interpreted parameters is higher than with the conventional monitor. In addition, we assume that the patient installations can be remembered more correctly since they are displayed on the avatar in the correct position for the patient and are thus perceived directly pictorially without processing any written information.

## 2. Materials and Methods

In this study, we analyze eye-tracking data which were collected during a multicenter, international, computer-based study [23]. In the five study centers, the University Hospital Zurich and the Hirslanden Clinic Zurich in Switzerland, the University Hospital Wuerzburg and the University Hospital Frankfurt in Germany, and the Hospital Clinic de Barcelona in Spain, we recruited five physicians and five nurses from each, working in an intensive care unit or holding an intensive care board certification. A total of 50 participants were thus enrolled from June 2021 to August 2021.

According to the local ethics committees in Zurich, Switzerland, Germany, and Spain, this study did not fall under the Human Rights Act, so no ethics approval was needed. Still, before commencing the study, we gained written informed consent from all participants to use their data anonymously for scientific evaluation. The participants took part in the study voluntarily and received no financial compensation.

### 2.1. Description of the Visual-Patient-Avatar Technology

Inspired by the synthetic vision technologies of aviation, the Visual-Patient-avatar was invented 10 years ago by our research group at the Institute of Anesthesiology, University and University Hospital Zurich, Switzerland and has been continuously refined since then, so that currently 15 vital signs and installations can be displayed [19,21]. In the background, the monitor data of the respective parameters are pre-processed into the categories ‘not measured’, ‘too low’, ‘normal’, or ‘too high’ and displayed as an avatar, which shows the alterations as a change in color, shape, or pulsation of the avatar’s elements. The goal is to display the data in real-time and as close to reality as possible, creating a virtual animation of a human being. Based on a user-centered design, the Visual-Patient avatar is intended to improve situation awareness in level 1 (perception), level 2 (comprehension), and level 3 (projection) [4]. For example, low saturation is visualized by a purple skin color or a high heart rate is displayed as a fast pulsation of the avatar body. In the last development step, the Visual-Patient avatar was adapted to the conditions of an intensive care unit. Additional parameters, such as cardiac output and ICP (intracranial pressure), which are commonly used in an intensive care setting, were implemented. Further, the avatar now displays various installations, such as airway tubes, peripheral and central venous lines, or urinary catheter according to their precise location in the patient. Figure 1 shows all currently possible vital signs and installations.

### 2.2. Study Design

This analysis is part of a prospective, multicenter, computer-based study comparing two different monitor modalities using eye-tracking technology in ICU patient scenarios [23]. Initially, we showed the participants an introductory video (Appendix A), in which the individual vital parameters and installations of the Visual-Patient-avatar ICU were displayed and explained in all possible states. The conventional monitor used in this study does not represent patient installations. This information is usually listed in writing in another program, often visible on the computer away from the patient’s bed. Therefore, the patient installations have been included below the conventional monitor so that both modalities can be compared. Thereafter, we showed each participant five cases in both modalities, resulting in 10 scenarios. We showed the 15 s videos in randomized order (Research Randomizer V4.0) in PowerPoint (Microsoft Corporation, Redmond, Washington, USA) on a laptop (Apple Inc., Cupertino, CA, USA). Appendix A [23] illustrates all cases in both modalities and the survey. The participants’ task was to remember if the vital parameters displayed in the videos were either “too low”, “too high”, or “normal”. Additionally, the participants had to remember the installations in a patient as well as their location. The answers were collected after each scenario using an iPad (Apple Inc., Cupertino, CA, USA) data collection tool (Harvest your data, Wellington, New Zealand presented in the app iSurvey) where the answers could be given by ticking boxes. The participants were also able to tick a box with “no recall”, in case they did not remember if the specific vital sign or installation was present or in which direction the deviation was. After each scenario, participants reported their subjective diagnostic confidence and perceived workload as determined by the NASA (National Aeronautics and Space Administration) Task Load Index [30,31]. At the end of the study, participants also had the opportunity to provide feedback on the Visual-Patient-avatar ICU technology and the study process.

### 2.3. Data Collection and Analysis

Eye-tracking data, specifically visual fixations and saccades, were recorded by a stationary eye tracker (Gazepoint GP3, Gazept, Vancouver, BC, Canada) placed at the bottom of the screen. The eye tracker was recalibrated for each participant before the start of the study and the position of the foveal vision on the screen was recorded 60 times per second with 0.5 to 1.0 degree of visual angle accuracy.

For post-hoc analysis, we used the Gazepoint Professional Analysis software (Gazepoint GP3, Gazept, Vancouver, BC, Canada) on an Acer Aspire V15 Nitro laptop (Acer Inc., Taipei, Taiwan). Before the videos were analyzed semiautomatically, they were independently reviewed for quality by two authors (JV and SA) and excluded if judged by consensus to be of low quality, which was defined as the gaze being outside the monitor more than half of the time. Within the software, we created specific Areas of Interest (AOIs) for each case and the two monitor modalities. As each of the cases shown had a different number of vital signs and installations, the different scenarios have a varying number of AOIs. The fixations and their durations within these AOIs were automatically exported by the Gazepoint Analysis software (Gazepoint GP3, Gazept) and inserted into a Microsoft Excel sheet (Microsoft Corporation) for further analysis.

An exemplary analysis with drawn AOIs for both modalities in the Gazepoint software can be found in Figure 2.

### 2.4. Outcomes and Statistical Analysis

#### 2.4.1. Outcomes

As a primary endpoint, we examined which Areas of Interest of each monitor modality were viewed most frequently and for the longest time. In addition, we investigated whether there was an association between the length of time spent viewing a specific Area of Interest and correctly solving the associated questions.

#### 2.4.2. Statistical Analysis

All statistical analyses were performed in R-studio (R Core Team (2022), Version 4.2.0, Vienna, Austria). For descriptive statistics, we show means, standard deviations, medians, interquartile ranges, minimum and maxim for continuous variables, and numbers and percentages for categorical variables.

To examine the influence of the time spent looking at an area on giving a correct answer, we calculated mixed logistic regression models with a random intercept per participant to take into account the fact that the questions answered by the same person were not independent. We additionally included a random intercept for each question to cover the differing difficulty of the questions. Apart from the modality variable, the model was adjusted for the respective case and the time spent looking at each area of interest.

Due to a high percentage of missing values, multiple imputation by chained equations with 100 imputed data sets was performed as a sensitivity analysis. The influence of time spent looking at an alarm on the frequency of correctly answered questions was analyzed with a linear mixed model with a random intercept per participant, adjusted for the respective case.

## 3. Results

### 3.1. Study and Participant Characteristics

We recruited 10 participants per study center, leading to a total of 50 participants, where 25 (50%) were physicians and 25 (50%) were nurses. Detailed information on participant characteristics can be found in the main study [23]. Each participant completed 10 scenarios, resulting in a total of 500 scenarios being analyzed. A total of 19 (38%) of the participants were female and the median age was 37.0 years (Interquartile Range (IQR) 33.0–43.8 [Min/Max 27–56]). For the analysis of the eye tracking data, 55% of the data points were missing. Missing data were mainly due to pre-existing vision correction with glasses (26%), interfering with the recording of the eye-tracking data. The second most common reason for missing data was technical problems at 19%. The remaining 10% of results from missing AOIs were because not all cases included all installations or vital signs.

The odds of correctly answering the associated question were higher with the Visual-Patient-avatar ICU compared with the conventional monitor (Odds Ratio [OR] 1.70, 95% CI 1.57–1.83, *p* < 0.001). This was confirmed in the sensitivity analysis with multiple imputations (OR 2.07, 95% CI 1.79–2.39, *p* < 0.001).

### 3.2. Areas of Interest

The Areas of Interest differ fundamentally between the two modalities and the individual cases in number, type, and size. While nine AOIs were defined for the Visual-Patient-avatar ICU and a maximum of seven of them were present in Case 1 for example, there are fifteen different AOIs for the conventional monitor, with all of them being shown in Case 2. Because of these differences, we describe the time spent on the AOIs in purely descriptive terms. With the conventional monitor, the participants looked at the installation site for the longest time on average (Median 2.13–2.51 s, depending on the case). In scenarios with the Visual-Patient avatar, the temporal distribution of the different areas was balanced, so that no area of interest was viewed for a particularly long time. Figure 3 shows, for both monitor modalities, how long participants looked at the different Areas of Interest per case. In summary, the longer an area was looked at, the more likely the associated question was answered incorrectly (OR 0.97, 95% CI 0.95–0.99, *p* = 0.008).

### 3.3. Alarms

In the upper right part of the conventional monitor, alarm messages are displayed in the event of deviations of the vital parameters from the set limit values. This area was not present in the Visual-Patient-avatar ICU. Depending on the severity of the deviation from the limit values, the blinking alarm messages appear either yellow for a slight deviation or red in case of a serious deviation. Although there were no associated questions for the alarm notifications in the questionnaire, the eye-tracking data for this area of the monitor were nevertheless collected and analyzed. An ANOVA test revealed no evidence of differences in time spent on alarm region between cases (*p* = 0.193), although cases 2 (median 0.11 s), 3 (0.18 s), and 4 (0.22 s), which contained both yellow and red alarms, were considered longer than case 1 (median 0.00 s) and case 5 (median 0.02 s). However, a closer examination of the different alarm types shows moderate evidence that more time was spent on alarms when both yellow and red alarms were present (*p* = 0.018). Overall, there was no evidence of an influence of the time spent looking at the alarms on the frequency of correctly answered questions (estimate 0.15, 95% CI −1.11–1.42, *p* = 0.818). The results are shown graphically in Figure 4.

## 4. Discussion

This study evaluated eye-tracking data from 50 participants collected during assessing five patient cases, either with the conventional monitor or with the Visual-Patient-avatar ICU. After each scenario, participants had to indicate whether and in what direction various vital signs were altered and what installations the patient had. When performing the scenarios with the Visual-Patient-avatar ICU, the participants recaptured more information correctly with higher diagnostic confidence and lower perceived workload compared to the conventional monitor [23]. In this study, we investigated how long the participants looked at individual areas of the monitor and how the influence of time spent on the different areas was on the correct answering of the questions. The main results show that there is no statistical significance in the Visual-Patient-avatar ICU between the different AOIs in terms of time spent on it. In contrast, with the conventional monitor, the area where the installation location is listed was viewed the longest. Although this specific area of the monitor was looked at for a particularly long time, the associated questions about the location of the installations were answered incorrectly significantly more often with the conventional monitor (RR 1.57, 95% CI 1.45–1.71, *p* = 0.03) [23]. According to this, the evaluation showed that the longer an area was observed, the more likely was the corresponding question answered incorrectly, taking into account the fact that this result was influenced mainly by the AOI ‘installation place’. The reason why the information on the installation location was viewed for a particularly long time on the conventional monitor could be that this information was available in written form and had to be read carefully by the participants, which required longer viewing. In addition, to the best of our knowledge, it is unusual for the installations to be displayed on the patient monitor, even though this was done here for study purposes. It can therefore not be ruled out that this and the choice of font size has an effect on the results. The additional cognitive challenge in the form of written information can lead to a decline in performance, especially in a high-pressure working environment and gives reason to present medical information as simply as possible [32]. This observation is also consistent with the fact that written information is generally more difficult to perceive than pictorial information, also known as the picture superiority effect and first described by Nelson et al. in 1976 [33]. The picture superiority effect describes the advantage of pictorial over written information, as no cognitive translation of the writing needs to be conducted, thus allowing for a direct logical interpretation of the information shown. If the pictorial information is furthermore presented according to logical principles and user-centered design, this can lead to a facilitation of situation awareness [4,22]. The Visual-Patient avatar makes use of these effects and was developed more than 10 years ago with the intention of improving the transfer of information and thus reducing cognitive load [19]. Various computer and simulation studies have shown in the past that with the Visual-Patient avatar, the information presented was perceived more quickly and correctly and the therapeutic actions derived from it were carried out more accurately, with a lower perceived subjective workload [21]. With the previous study, we were also able to validate these results for the Visual-Patient-avatar ICU [23]. The recently enhanced version, which in addition to the vital signs also displays the patient’s installations, according to their place on the avatar, is intended to further promote situation awareness by presenting important information pictorially and in a single display [34].

This study shows that with the Visual-Patient-avatar ICU, the gaze on the monitor is more balanced over all individual areas than with the conventional monitor. In particular, written information takes much more time to be perceived as a surrogate parameter for cognitive performance. This supports our effort to present medical information with visualizations in a simple and intuitive way. As the Visual-Patient-avatar ICU has only been tested in a study setting during its development, we would now like to move on to the next step. The Visual-Patient-avatar ICU will be introduced and further developed in the context of an already planned real-life introduction into daily clinical work. The avatar will be displayed on the screen as an add-on to the conventional monitor.

### Limitations and Strengths

This study has several limitations. The data we analyzed in this study were collected as part of another study. Therefore, no sample size calculation was performed for the purpose of this study and no general statement can be made, especially considering that much data had to be excluded due to insufficient eye-tracking quality. The results could differ in both directions with more data points. The interpretation of eye-tracking data is limited and does not take into account other factors such as working memory or peripheral vision which may influence the perception of information [29]. On the other hand, there is the fact that in the past, a positive correlation between visual fixation and correct perception could be shown [29]. The avatar-based visualization of the patient’s condition can be perceived significantly better compared to the conventional monitor with peripheral vision [17,35]. Since the configurations of the two monitors are very different, the individual AOIs cannot be directly compared with each other as they contain different vital signs and installations. To account for this, we defined which vital signs or installations were shown for each area of interest individually. The strengths of this study include the multicenter and international design and the balanced participant selection. Thus, regional influences on the results can be excluded. The purely computer-based study design is particularly well-suited for the evaluation of eye-tracking data since no distraction and interference factors are added.

## 5. Conclusions

The evaluation of the eye-tracking data showed that the Visual-Patient-avatar ICU facilitates and improves information transfer through its visualizations. The different areas of the monitor were viewed in a balanced way with the Visual-Patient-avatar ICU. Whereas with the conventional monitor, although the most time was spent on the area where the installation location was written, the related questions were more often answered incorrectly. For both modalities, the longer an area was viewed, the more likely the associated question was answered incorrectly. In our view, in line with the picture superiority effect, this reflects an increased cognitive effort for the perception of written information and indicates that visualizations should be further encouraged in order to present medical information as simply as possible in clinical routines [33].

## Figures and Tables

**Figure 1 diagnostics-13-03432-f001:**
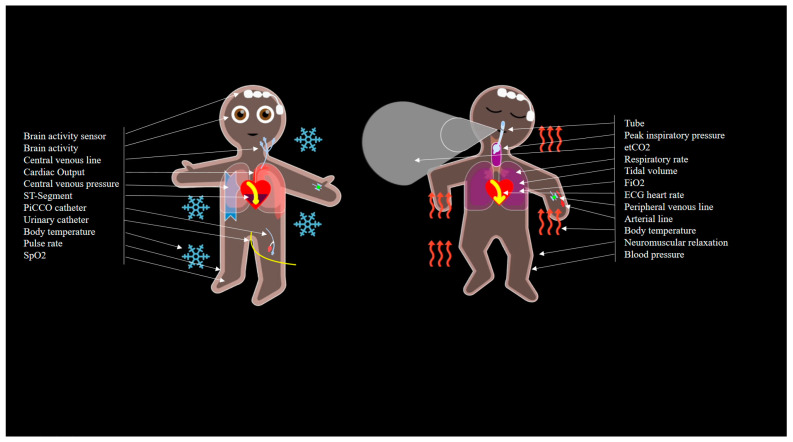
Presented in two examples are the currently available vital signs and installations in the Visual-Patient-avatar ICU. SpO2: peripheral oxygen saturation. etCO2: end-tidal carbon dioxide. FiO2: inspiratory oxygen concentration. ECG: electrocardiogram.

**Figure 2 diagnostics-13-03432-f002:**
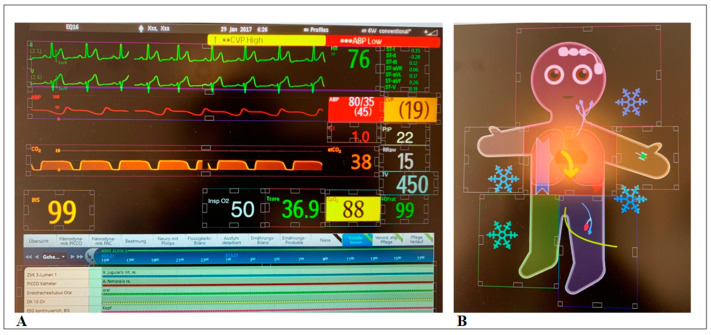
Example of the Areas of Interest drawn in both modalities (conventional monitor on the left (**A**), Visual-Patient-avatar ICU on the right (**B**)) in Gazepoint Analysis. The colored squares and rectangles symbolize individual areas of interest within which the gaze data are collected.

**Figure 3 diagnostics-13-03432-f003:**
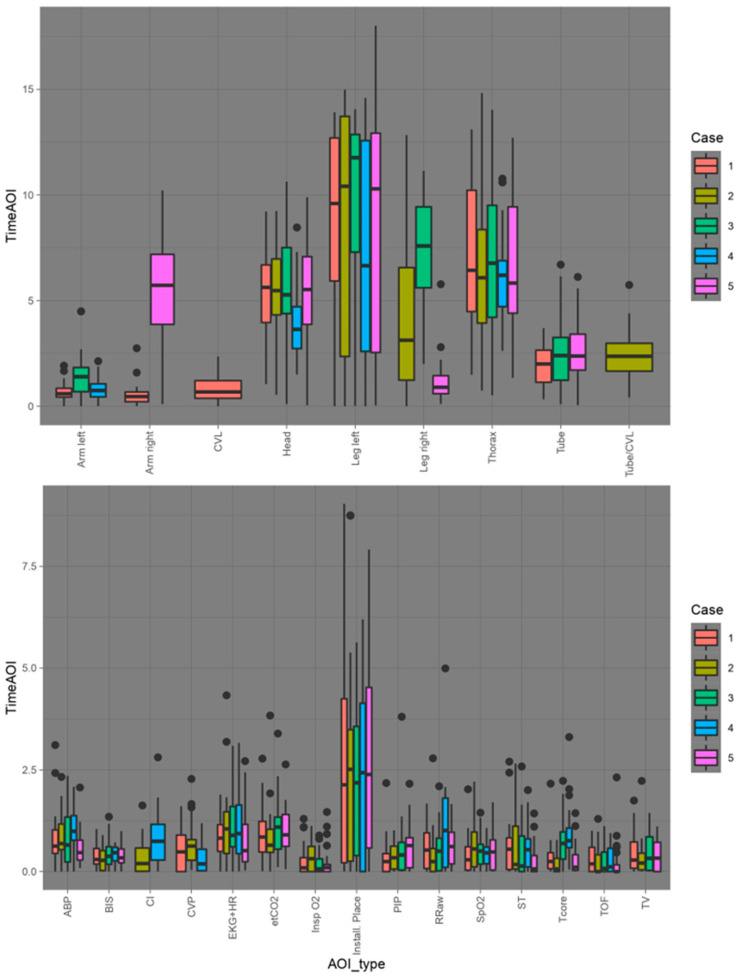
Graphical representation of the duration of viewing the individual Areas of Interest (AOI) in seconds. The upper image shows the AOIs of the Visual-Patient-avatar ICU. In the lower image are the AOIs of the conventional monitor. The different colors correspond to the different cases. CVL = Central venous line. ABP = Arterial blood pressure. CI = Cardiac Index. CVP = Central venous pressure. ECG = Electrocardiogram. HR = Heart rate. PIP = Peak inspiratory pressure. Tcore = Core temperature. TOF = Train of four. TV = Tidal volume.

**Figure 4 diagnostics-13-03432-f004:**
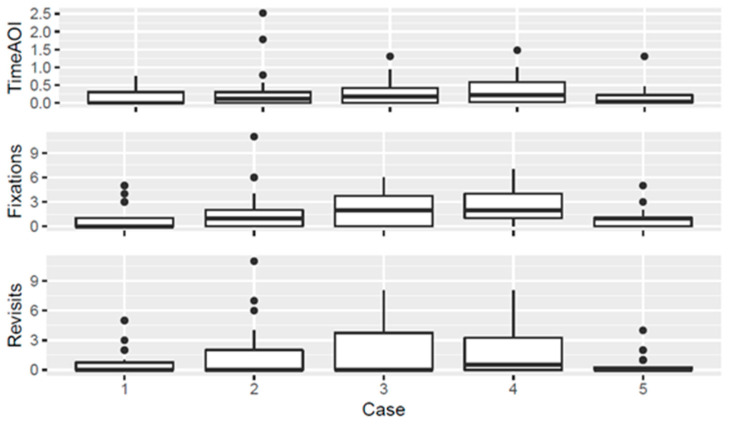
Graphical representation of time spent on alarm regions, with the *X*-axis showing cases 1–5 and the *Y*-axis showing time in seconds for TimeAOI (**top**), and numerical count for Fixations (**middle**) and Revisits (**bottom**).

## Data Availability

The datasets used and/or analyzed during the current study are available from the corresponding author on reasonable request.

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
