# Peer review of "The Visual Patient Avatar ICU Facilitates Information Transfer of Written Information by Visualization: A Multicenter Comparative Eye-Tracking Study"

_diagnostics, 2023, doi:10.3390/diagnostics13223432_

Round 1

Reviewer 1 Report

There is nothing to criticise about the concept of the present study: Different professional groups (doctors and nurses) from different clinics were included, the number of test series is well dimensioned (50 x 10 randomised scenarios), and the technical equipment (Gazepoint GP3) is considered sufficiently accurate and reliable for the recording of eye movements.

However, the postulated correlation between the time spent on an AOI and the error rate in answering the corresponding questions should be corrected in two points:

(1) Lines 211-213 and 254-255 describe that the Visual-Patient Avatar showed no statistically significant differences in terms of the durations over which the AOIs were viewed. How then can there be a correlation between viewing time and error rate in answering the corresponding questions? (Line 215: "it applies to both monitor modalities, the longer an area was looked at, the more likely the associated question was answered incorrectly"). There is a need for clarification here.

(2) In the conventional monitor group, the length of stay at the "installations" indeed differs significantly from all other AOIs, and questions on this item were also answered incorrectly more frequently. However, a causality between length of stay and error rate (line 260: "...the longer an area was observed, the more likely was the corresponding question answered incorrectly") can hardly be deduced from this. A very simple and plausible common reason for both (!) phenomena could be, that the information on the installations is written in microscopically small letters, which is why they take longer to record and are subject to more errors. In addition, it might be absolutely unusual for most users to look for information about i.v. accesses etc. on a "classic" patient monitor.

In my opinion, the discussion should therefore be limited to the conclusion that the more complex the information is presented, the longer it takes to gather information and the higher the error rate. All conclusions that go beyond this are speculative.

Finally, the legend of Figure 3 seems to be written in German. This should be translated into English.

Author Response

There is nothing to criticise about the concept of the present study: Different professional groups (doctors and nurses) from different clinics were included, the number of test series is well dimensioned (50 x 10 randomised scenarios), and the technical equipment (Gazepoint GP3) is considered sufficiently accurate and reliable for the recording of eye movements.

However, the postulated correlation between the time spent on an AOI and the error rate in answering the corresponding questions should be corrected in two points:

  1. Lines 211-213 and 254-255 describe that the Visual-Patient Avatar showed no statistically significant differences in terms of the durations over which the AOIs were viewed. How then can there be a correlation between viewing time and error rate in answering the corresponding questions? (Line 215: "it applies to both monitor modalities, the longer an area was looked at, the more likely the associated question was answered incorrectly"). There is a need for clarification here.
  • Reply: Thank you very much for your kind revision and helpful input. It is indeed the case that in the visual-patient-avatar ICU comparison, the areas of interest do not show a significant difference in time. However, if one takes all times of all Areas of Interest together and puts them in correlation with the incorrectly answered questions, a correlation can be derived, which states that when a specific Area of Interest was viewed for a longer time, the associated question was more likely to be answered incorrectly. This correlation is mainly due to the results of the conventional monitor. To bring this out better, we have changed this accordingly.
  • Changes:
    • Line 215-217: In summary of all findings, it applies, the longer an area was looked at, the more likely the associated question was answered incorrectly (OR 0.97, 95% CI 0.95 – 0.99, p=0.008).

  1. In the conventional monitor group, the length of stay at the "installations" indeed differs significantly from all other AOIs, and questions on this item were also answered incorrectly more frequently. However, a causality between length of stay and error rate (line 260: "...the longer an area was observed, the more likely was the corresponding question answered incorrectly") can hardly be deduced from this. A very simple and plausible common reason for both (!) phenomena could be, that the information on the installations is written in microscopically small letters, which is why they take longer to record and are subject to more errors. In addition, it might be absolutely unusual for most users to look for information about i.v. accesses etc. on a "classic" patient monitor.
  • Reply: Best thanks for these considerations. It is indeed from the statistical calculations that the AOI 'installations' was viewed the longest for the conventional monitor. Moreover, the correlation of the duration of viewing and incorrect answering of the related question is statistically evaluated correctly. However, this fact is mainly influenced by the AOI 'installations'. We have added this in our discussion for better understanding. We agree, it is indeed very uncommon for patient installations to be displayed on a monitor. We solved this in the study for comparability of data (compare Study design, line 125-129). However, we agree with you: possibly a larger font size would have had an influence on the results. Even though we did not receive any corresponding feedback from the participants. We have added this to the paper accordingly.
  • Changes:
    • Line 262-265: According to this, the evaluation showed that the longer an area was observed, the more likely was the corresponding question answered incorrectly, taking into account the fact that this result was influenced mainly by the AOI ‘installation place’.
    • Line 268-271: In addition, to the best of our knowledge, it is unusual for the installations to be displayed on the patient monitor, even though this was done here for study purposes. It can therefore not be ruled out that this and the choice of font size has an effect on the results.

  1. In my opinion, the discussion should therefore be limited to the conclusion that the more complex the information is presented, the longer it takes to gather information and the higher the error rate. All conclusions that go beyond this are speculative.
  • Reply: Thank you very much for this input. We can show through our study that the viewing of the individual areas of interest was balanced in the Visual-Patient-avatar ICU, whereas in contrast the written information took a lot of time in the conventional monitor. This is a purely descriptive result, which we find very interesting and would like to leave as it corresponds to our vision of the simplest possible presentation of medical data. We have stated this in the last paragraph of the discussion.
  • Changes: none

  1. Finally, the legend of Figure 3 seems to be written in German. This should be translated into English.
  • Reply: We translated legend of figure 3.
  • Changes:
    • Line 219-222: Graphical representation of the duration of viewing the individual Areas of Interest (AOI) in seconds. The upper image shows the AOIs of the visual patient avatar ICU. In the lower image, the AOIs of the Conventional Monitor. The different colors correspond to the different cases.

Reviewer 2 Report

The article developed by your team is interesting even from the introduction and keeps the reader's interest awake as it is read.

The method used for monitoring based on eye tracking and visualization of the area of interest is innovative and significantly simplifies the activity in the intensive care service.

Of course, there are still many steps to be taken to implement this system in intensive care services, but through the study presented in the manuscript sent for publication, there are promising perspectives.

I believe that the study is well conducted, it illustrates some pertinent results and appropriate discussions, but it requires reworking the conclusions so that it properly reflects both the current state and the possibility of implementation in daily clinical practice. In addition, it would be necessary to correct the text related to Figure 3 so that it is translated from German to English, considering that the rest of the manuscript is in English.

Author Response

The article developed by our team is interesting even from the introduction and keeps the reader's interest awake as it is read.

The method used for monitoring based on eye tracking and visualization of the area of interest is innovative and significantly simplifies the activity in the intensive care service.

Of course, there are still many steps to be taken to implement this system in intensive care services, but through the study presented in the manuscript sent for publication, there are promising perspectives.

I believe that the study is well conducted, it illustrates some pertinent results and appropriate discussions, but it requires reworking the conclusions so that it properly reflects both the current state and the possibility of implementation in daily clinical practice. In addition, it would be necessary to correct the text related to Figure 3 so that it is translated from German to English, considering that the rest of the manuscript is in English.

  • Reply: Thank you very much for your favorable evaluation of our paper and the valuable input. We will gladly implement them and comment on the individual points.
  • Visual-Patient-avatar ICU has only been tested for study purposes in the course of its development. In the next, for us big, step the real-life introduction is planned. For this purpose, the Visual-Patient-avatar ICU will be displayed in the form of a split-screen next to the conventional monitoring on the screen. This study is already underway and we are looking forward to further development of Visual-Patient-avatar ICU. We have adjusted the last paragraph of the discussion accordingly. The legend to Figure 3 has been translated into English.
  • Changes:
    • We translated legend of figure 3.
    • Line 295-299: After Visual-Patient-avatar ICU has only been tested in a study setting during its devel-opment, we would now like to move on to the next step. Visual-Patient-avatar ICU will be introduced and further developed in the context of an already planned real-life introduc-tion in the daily clinical work. The avatar will be displayed on the screen as an add-on to the conventional monitor.

Reviewer 3 Report

I reviewed the article by Viautour et al, entitled “The Visual-Patient-avatar ICU Facilitates Information Transfer of Written Information by Visualization: a Multicenter Comparative Eye-tracking Study” (Manuscript ID: diagnostics-2604662) submitted to Diagnostics. Using the eye-tracking data, the authors mainly investigated which areas of interest were viewed most frequently and for the longest time between conventional monitor and visual patient avatar ICU monitor.

They found that the installation site was seen in the longest time in the conventional monitor, while gaze distribution was balanced in the visual-Patient avatar ICU. They also observed that he longer an area was viewed, the more likely the associated question was answered incorrectly.

From these observations, they claimed the usefulness of the Visual-Patient-avatar ICU.

First, the reviewer pays respect for the Authors' tremendous effort spent on this manuscript. However, there are several concerns with the data presentation, and the experimental design. My concerns are listed below:

Methods

1

How to develop the used survey questionnaire? The process of the survey questionnaire development should be described more in details.

2

Are measured outcomes (The Areas of Interest etc) are clinically relevant and why? This reviewer is wondering

3

Are the differences of measured outcome between two groups are clinically meaningful difference and why? The just few second difference on the monitor really hider the correct interpretation of the vital sings? This reviewer feels it is very unlikely.

4

Are there any potential confounders and effect modifiers in this study? If present, please clearly define and give strategy how to adjust for them.

5

Who planned this study, who enrolled the participants, who collected data, and who conducted the statistical analysis? I think if the same researchers are involved in study planning, data collecting, outcome measurement, and statistical analysis, there is a theoretical risk of biased assessment. Describe any efforts to address potential sources of bias. For example, blinding is one of the attractive methods to reduce above mentioned biased assessment. If done, please provide who was blinded and how.

Results

6

Characteristics of 50 participants (eg, clinical experience, specialties, board certification, etc) and characteristics of five cases (eg. etiology, age, sex, etc) should be described more in details, ideally using a table.

7

As the authors indicated, one of the strong limitations of this study is nearly 50% data are missing. This reviewer is wondering these missing data are Missing At Random or Missing Not At Random.

8

Figure 3 is too elaborated and complicated, and difficult to follow.

9

This manuscript contains large numbers of careless mistakes. For example, the legend of Figure 3 is written in German language. There are some other mistakes. Please check.

10

Discuss the generalizability (external validity) of the study results.

11

The author claimed that the usefulness of the Visual-Patient-avatar ICU

Although the number of criticisms listed above, this reviewer should however state that it is laudable that this work is derived from huge efforts made by the authors, who are working as the frontline healthcare professionals. The reviewer respects the authors time and effort spent on this manuscript, and the authors ‘patience and professionalism in dealing with my comments.

The author should consider to use the professional English editting service. 

Author Response

I reviewed the article by Viautour et al, entitled “The Visual-Patient-avatar ICU Facilitates Information Transfer of Written Information by Visualization: a Multicenter Comparative Eye-tracking Study” (Manuscript ID: diagnostics-2604662) submitted to Diagnostics. Using the eye-tracking data, the authors mainly investigated which areas of interest were viewed most frequently and for the longest time between conventional monitor and visual patient avatar ICU monitor.

They found that the installation site was seen in the longest time in the conventional monitor, while gaze distribution was balanced in the visual-Patient avatar ICU. They also observed that he longer an area was viewed, the more likely the associated question was answered incorrectly.

From these observations, they claimed the usefulness of the Visual-Patient-avatar ICU.

First, the reviewer pays respect for the Authors' tremendous effort spent on this manuscript. However, there are several concerns with the data presentation, and the experimental design. My concerns are listed below:

  1. How to develop the used survey questionnaire? The process of the survey questionnaire development should be described more in details.
  • Reply: Thank you very much for the thoughtful evaluation of our paper and the valuable input, which we will be happy to implement in order to improve our work. We are happy to comment on the individual points.

We describe the process of the study including the structure of the survey in our methodology section ‘2.2 study design’. The survey was created with the help of the software Harvest your data. After each 15-second video, participants received an iPad with the survey. For each vital sign, they had to indicate whether it was 'too low', 'too high', or 'normal'. This could be checked by simply clicking a box. In addition, participants indicated which installations could be seen and where they were located. If the participants could not remember what the state of a vital sign was and if so, where which installations were located, they could select 'no recall'. The survey is shown in Additional File 2. We have added this in the paper for better understanding.

  • Changes:
    • Line 132-133: Additional file 2 illustrates all cases in both modalities and the survey.

  1. Are measured outcomes (The Areas of Interest etc) are clinically relevant and why? This reviewer is wondering
  • Reply: We defined the Areas of Interest as specific areas of the monitor and were able to show that written information in particular takes more time to be perceived and processed. In addition, the associated questions were answered incorrectly more often. This may have clinical relevance, as working in an intensive care unit is cognitively very challenging. We therefore aim to present information as simply as possible in order to reduce cognitive load, as this has been shown to lead to more errors in clinical work.
  • Changes: none

  1. Are the differences of measured outcome between two groups are clinically meaningful difference and why? The just few second difference on the monitor really hider the correct interpretation of the vital sings? This reviewer feels it is very unlikely.
  • Reply: Thank you very much for this input. We agree with you: in the context of 15-second videos, the differences are in the range of seconds and therefore minimal. We have not investigated here the impact on the work in the clinical routine, where the monitor is viewed regularly and recurrently. However, this will be the case in the next step of further development of Visual-Patient-avatar ICU, the real-life introduction. However, we can state that the written information on the conventional monitor was viewed significantly longer with consequently more frequent incorrect answers to the corresponding questions. However, it must be considered that, according to our knowledge, it is unusual to find the information about the installations on the patient monitor. In addition, we cannot exclude an effect of the selected font size, even if this was not explicitly reported by the participants. We have added this accordingly in our paper.
  • Changes:
    • Line 268-271: In addition, to the best of our knowledge, it is unusual for the installations to be displayed on the patient monitor, even though this was done here for study purposes. It can therefore not be ruled out that this and the choice of font size has an effect on the results.

  1. Are there any potential confounders and effect modifiers in this study? If present, please clearly define and give strategy how to adjust for them.
  • Reply: We tried to eliminate all possible confounding factors. Participants were undisturbed during the study, cases were all shown on the same laptop, and videos were preprogrammed to 15 seconds. Due to the multicenter international design of the study, we hope that we were able to resolve local confounders.

  1. Who planned this study, who enrolled the participants, who collected data, and who conducted the statistical analysis? I think if the same researchers are involved in study planning, data collecting, outcome measurement, and statistical analysis, there is a theoretical risk of biased assessment. Describe any efforts to address potential sources of bias. For example, blinding is one of the attractive methods to reduce above mentioned biased assessment. If done, please provide who was blinded and how.
  • Reply: The study was designed, planned and conducted by the team of authors. The exact breakdown of the individual study steps is listed in the chapter 'Authors contributions' at the end of the article. Since the study team was present during the study procedure, blinding could not be performed. Thus, bias cannot be completely excluded, although we can assure you that the study participants were not influenced by us.

  1. Characteristics of 50 participants (eg, clinical experience, specialties, board certification, etc) and characteristics of five cases (eg. etiology, age, sex, etc) should be described more in details, ideally using a table.
  • Reply: Thank you very much for this input. This paper is based on the study ‘Bergauer L, Braun J, Roche TR, Meybohm P, Hottenrott S, Zacharowski K, Raimann FJ, Rivas E, López-Baamonde M, Ganter MT, Nöthiger CB, Spahn DR, Tscholl DW, Akbas S. Avatar-based patient monitoring improves information transfer, diagnostic confidence and reduces perceived workload in intensive care units: computer-based, multicentre comparison study. Sci Rep. 2023 Apr 11;13(1):5908. doi: 10.1038/s41598-023-33027-z. PMID: 37041316; PMCID: PMC10088750.’. Participant characteristics are discussed in detail there. We are happy to add a corresponding note in our paper.
  • Changes:
    • Line 192-193: Detailed information on participant characteristics can be found in the main study (23).

  1. As the authors indicated, one of the strong limitations of this study is nearly 50% data are missing. This reviewer is wondering these missing data are Missing At Random or Missing Not At Random.
  • Reply: We agree, this is indeed a limitation of our study. As we describe in the section 'Results' Line 196-200, 26% of the missing data points are due to the wearing of corrective glasses, which interferes with the eye tracker. Another reason was technical problems despite previous calibration.

  1. Figure 3 is too elaborated and complicated, and difficult to follow.
  • Reply: Thank you very much for this comment. We have thought about Figure 3 again in detail and have also consulted our statisticians again. In order to present the information that is important to us, we would like to stay with the current version of the Figure.

  1. This manuscript contains large numbers of careless mistakes. For example, the legend of Figure 3 is written in German language. There are some other mistakes. Please check.
  • Reply: Despite careful checking, we have unfortunately made spelling mistakes. We have translated the legend of Figure 3 and spell-checked the paper again.

  1. Discuss the generalizability (external validity) of the study results.
  • Reply: We can say with our study that the participants looked at the different areas of the monitor for different lengths of time. In the case of the visual-patient-avatar ICU, this was balanced; in the case of the conventional monitor, the area of the installations was viewed the longest. In addition, the associated questions were answered incorrectly more frequently. This suggests that taking in and processing written information requires increased cognitive work. Conversely, it is the case that pictorially presented information is easier to perceive. This corresponds to the picture superiority effect already statistically established by Nelson et al in 1976. Through our multicenter international study design, we were also able to eliminate local factors and make a certain generally valid statement. This is a very important result for us, as we aim to simplify the presentation of medical information in the daily, cognitively very challenging work in the intensive care unit.

Although the number of criticisms listed above, this reviewer should however state that it is laudable that this work is derived from huge efforts made by the authors, who are working as the frontline healthcare professionals. The reviewer respects the authors’ time and effort spent on this manuscript, and the authors ‘patience and professionalism in dealing with my comments.

Round 2

Reviewer 3 Report

I have no remaining comments regarding this manuscript.